# *KCND3*-Related Neurological Disorders: From Old to Emerging Clinical Phenotypes

**DOI:** 10.3390/ijms21165802

**Published:** 2020-08-13

**Authors:** Luca Pollini, Serena Galosi, Manuela Tolve, Caterina Caputi, Carla Carducci, Antonio Angeloni, Vincenzo Leuzzi

**Affiliations:** 1Department of Human Neuroscience, Sapienza University of Rome, 00185 Rome, Italy; luca.pollini@uniroma1.it (L.P.); serena.galosi@uniroma1.it (S.G.); caterina.caputi@uniroma1.it (C.C.); 2Department of Experimental Medicine, Sapienza University of Rome, 00185 Rome, Italy; manuelatolve@gmail.com (M.T.); carla.carducci@uniroma1.it (C.C.); antonio.angeloni@uniroma1.it (A.A.)

**Keywords:** KCND3, SCA19/22, ataxia, Kv4.3, neurodevelopmental disorder, cerebellum, movement disorders, parkinsonism, dystonia

## Abstract

*KCND3* encodes the voltage-gated potassium ion channel subfamily D member 3, a six trans-membrane protein (Kv4.3), involved in the transient outward K^+^ current. *KCND3* defect causes both cardiological and neurological syndromes. From a neurological perspective, Kv4.3 defect has been associated to SCA type 19/22, a complex neurological disorder encompassing a wide spectrum of clinical features beside ataxia. To better define the phenotypic spectrum and course of *KCND3*-related neurological disorder, we review the clinical presentation and evolution in 68 reported cases. We delineated two main clinical phenotypes according to the age of onset. Neurodevelopmental disorder with epilepsy and/or movement disorders with ataxia later in the disease course characterized the early onset forms, while a prominent ataxic syndrome with possible cognitive decline, movement disorders, and peripheral neuropathy were observed in the late onset forms. Furthermore, we described a 37-year-old patient with a de novo *KCND3* variant [c.901T>C (p.Ser301Pro)], previously reported in dbSNP as rs79821338, and a clinical phenotype paradigmatic of the early onset forms with neurodevelopmental disorder, epilepsy, parkinsonism-dystonia, and ataxia in adulthood, further expanding the clinical spectrum of this condition.

## 1. Introduction 

Voltage-gated potassium ion channels (Kvs), considered highly sophisticated voltage-gated ion channels from both a functional and a structural standpoint, play a fundamental role in the generation and propagation of the action potential in all living beings [1]. Kvs are involved in a large variety of biological processes including neurotransmitter release, neuronal excitability, epithelial electrolyte transport, smooth muscle contraction, cell volume, and regulation of apoptosis [2,3]. Kv4 channels, including the Kv4.1, Kv4.2, and Kv4.3 subtypes, are a sub-group of the large Kvs family expressed in brain, heart, and smooth muscles [4]. Kv4.3 or Potassium Voltage-Gated Channel Subfamily D Member 3, encoded by *KCND3*, is a six trans-membrane segmented (S1–S6) ion channel, involved in the transient outward K^+^ current. Segments S1 to S4 form the voltage-sensing domain, while S5 and S6 segments with the pore loop constitute the ion selective pore (H5) [5]. 

Kv4.3 is highly expressed in the central nervous system, particularly in cerebellar Purkinje cells, deep nuclei, granule cells, and interneurons [6]. 

Pre- and postmigrating Purkinje cells show different levels of Kv4.3 expression, suggesting a role in their migration and cerebellar development [7]. 

*KCND3* is the causative gene of SCA 19/22, an autosomal dominant cerebellar ataxia mapped to chromosome 1p21-q23 [5,8].

This condition was first identified through linkage analysis in two independent large families [9,10] and named, respectively, spinocerebellar ataxia (SCA) type 19 [9] and type 22 [10]. Interestingly, according to the Harding’s clinical classification of autosomal dominant cerebellar ataxia (ADCA) [11], the SCA19 family was characterized as ADCA type 1 (Autosomal Dominant Cerebellar Ataxia with ophtalmoplegia, dementia, optic atrophy, pyramidal and extrapyramidal features) [9], while the SCA22 family was characterized as ADCA type 3 (“pure” cerebellar ataxia) [10]. More than 80 patients have been reported over time, broadening the clinical spectrum of *KCND3* deficiency to include epilepsy, intellectual disability/cognitive impairment, movement disorders, pyramidal signs, and peripheral neuropathy [5,6,8,9,10,12,13,14,15,16,17,18,19,20]. Furthermore, Kv4.3 defect has also been associated with cardiological phenotypes, including Brugada’s syndrome and early-onset atrial fibrillation [21]. Although pathogenesis is not fully understood, *KCND3* loss-of-function mutations have been associated with the neurological phenotype, while *KCND3* gain-of-function has been associated with the cardiological syndrome, with only few exceptions [12].

In order to better delineate the neurological phenotype associated with *KCND3* gene deficiency, we reviewed the literature, focusing on clinical presentation and evolution according to age of onset and highlighting less frequent and underrecognized clinical features of this condition. Finally, we report a new case presenting in childhood with neurodevelopmental disorder and focal epilepsy who later developed a complex movement disorder and, into adulthood, ataxic features and cerebellar atrophy.

## 2. Results

### 2.1. Case Report

This 37-year-old man was born from a non-consanguineous healthy couple. Delivery was at term with no complications. Early development was normal, with first word pronounced at 12 months of age and walking at 15-months. He was first evaluated at three years of age for speech difficulties and clumsiness with frequent falls. Rehabilitative treatment improved clumsiness and falls. At the age of 5 years, neuropsychological tests revealed a mild intellectual disability (IQ = 67) on the Weschler Intelligence Scale for Children III (WISC-III). On neurological examination, he presented minimal upper limbs postural tremor, brisk lower limbs reflexes and oral and verbal dyspraxia. During the same period, he suffered from both focal and generalized motor seizures. EEG recording detected frontocentral epileptic abnormalities with slow generalized background activity. Carbamazepine treatment successfully controlled epilepsy. At that time, brain MRI was normal. At the age of 7, a further cognitive evaluation showed a moderate intellectual disability (IQ = 41 on WISC-III) and behavioral dyscontrol became a major issue. 

At the age of 14 years, he started to complain about recurrent migraine episodes. In the same period, carbamazepine was withdrawn due to a long-standing seizure freedom. The next year he suffered from episodes of left-sided facial numbness, dizziness, sweating, limb and head tremors with migraine. He also progressively exhibited cervical and upper limbs dystonia, more severe during evening hours. A levodopa trial was ineffective. Clinical status remained relatively stable over the next years, and repeated brain MRI failed to detect any alteration. By his twenties he showed a more pronounced cervical and oromandibular dystonia, a subcontinuous “no-no” head tremor, bradykinesia, and generalized rigidity. A slightly wide-based gait became evident. A brain MRI at the age of 23 years revealed isolated vermis atrophy. An extensive work-up, including plasma aminoacids, acylcarnitine on dried blood spot, urinary organic acids, copper metabolism study, cerebrospinal fluid neurotransmitters and pterins, alfafetoprotein, and lysosomal enzyme activities, was normal. Two next generation sequencing panels for genetic epilepsy and genetic movement disorders did not reveal any responsible genetic alteration. Finally clinical exome sequencing (CES) analysis disclosed a de novo c.901T>C (p.Ser301Pro) *KCND3* variant (Figure 1), predicted deleterious by different tools (Mutation Taster [22], Predict SNP [23], HOPE [24], and Varsome [25]). This variant was reported in dbSNP (https://www.ncbi.nlm.nih.gov/snp/, last access date: 11 June 2020) as rs79821338 but no clinical description was available. The variant was located in the S4 functional domain and mutated aminoacid was predicted to alter physical and chemical properties and interactions between the transmembrane protein domain and the lipid membrane, disturbing the S4 domain function.

Neurological examination at the age of 37 revealed bradykinesia, rigidity, a mixed parkinsonian-ataxic gait, cervical and oromandibular dystonia with dystonic head tremor, dystonic posturing of the upper limbs (left > right), and cerebellar speech. He also exhibited saccadic ocular pursuit, saccadic dysmetria, and an ocular fixation instability (see Appendix A for neurological examination at the ages of 27 and 37). He was evaluated using the International Cooperative Ataxia Rating Scales (ICARS) and scored 23/100. 

### 2.2. Age at Onset, Presentation, and Disease Course

Age of onset ranged from the first year to the ninth decade of life [5,17]. Ataxia was the presenting sign in 42 out of 68 patients [5,6,8,9,12,14,16,18]. Five patients presented with epilepsy [16,19] and five others with neurodevelopmental disorder with or without cognitive impairment [6,9,13,17]. Less frequenti presenting symptoms were episodic ataxia (EA) (2 patients) [15,20] and head tremor (2 patients) [9,18], intermittent diplopia (1 patient) [5] and psychiatric symptoms [5]. 

Clinical course was characterized by the emergence over time of different patterns of neurological impairment. A pure cerebellar syndrome, defined as the presence of gait abnormalities, imbalance, dysmetria, dysdiadochokinesia, intentional tremor, cerebellar oculomotor disturbance (i.e., nystagmus, saccadic pursuit, slow saccades) and dysarthria was detected in 22 patients [5,6,8,9,14,16]. Other patients experienced non-cerebellar oculomotor disturbance (i.e., vertical ophthalmoplegia or supranuclear gaze palsy), cognitive impairment, intellectual disability (ID), movement disorders including parkinsonism, dystonia, myoclonus and tremor, epilepsy, pyramidal signs, or peripheral neuropathy in addition to ataxia and cerebellar oculomotor disorders [5,6,8,9,12,13,16,17,18,19,20]. Clinical features of reported patients are summarized in Table 1. 

### 2.3. Clinical Phenotype According to the Age of Onset

While ataxia was the most frequent presenting sign in late-onset patients, a variable association of ataxia, neurodevelopmental disorder (developmental delay followed by ID), and epilepsy characterized the presentation in the early-onset forms.

The clinical course in the early-onset cohort was characterized by neurodevelopmental disorders (15/15), ataxia (14/15), and oculomotor disorders (10/15) as main symptoms [6,9,13,16,17,18,19].

Epilepsy was reported exclusively in early-onset patients and occurred in nearly 50% of patients (7 out of 15 early-onset patients) [13,16,19]. Eight patients exhibited movement disorders [6,9,16,17,18]. Three out of 15 patients showed pyramidal signs [9,16,17].

A pure cerebellar syndrome characterized a relevant number of the late-onset patients (22/53) [5,6,8,9,14,16]. Possible adjunctive symptoms were movement disorders (15/31), pyramidal signs (13/31), cognitive impairment (13/31), and peripheral neuropathy (5/31) (Appendix A). Percentage of symptoms for the early- and the late-onset cohort are available in Figure 2.

### 2.4. Ataxia

Fixed and progressive ataxia, which was reported in nearly all patients (64/68), included a variable association of gait disturbance, limbs dysmetria, dysdiadochokinesia, cerebellar dysarthria, and swallowing disturbances. The course of ataxia in late-onset patients was usually slowly progressive, with patients bedridden or wheelchaired after 30–50 years of the disease [6,16]. In the first described family, the medium ICARS score of 8 patients was about 35/100 after 31 years of disease [9]. Two other individuals with uncertain duration of disease from the same family scored 2/100 and 6/100 at the age of 28 and 11, respectively [9]. Two patients evaluated with the Scale for the Assessment and Rating of Ataxia (SARA) scored 12/40 and 17/40 after 9 and 29 years of disease, respectively [6]. Four family members showed an increment of 0 to 6 points on SARA after 2 to 5 years [18]. Furthermore, Lee reported a very slow progression of 0.3 score point per year on SARA during a 5 years follow-up in one case, and an individual with a low score (9/40) after 31 years of disease duration [5]. In early-onset patients, ataxia had a less predictable course in terms of age of onset, severity, and progression rate [13,16].

### 2.5. Oculomotor Disorders

A proper oculomotor performance is strictly dependent on an intact cerebellum [26]; thus, it is not surprising that oculomotor disorders are one of the most frequently associated signs in *KCND3* deficiency (39/68 patients). Saccadic smooth pursuit and gaze-evoked-nystagmus (GEN), usually considered expressions of vestibulocerebellar dysfunction [27,28], were described in 27 and 16 individuals, respectively. Saccadic dysmetria was reported in three patients [9,18], described as hypometric saccades in two of them [18]. Slow eye saccades were reported once [6]. Late-onset patients showed other disturbances including diplopia (1), intermittent diplopia (1), downbeat nystagmus (1), and non-cerebellar oculomotor disorders such as vertical ophtalmoplegia (1) and supranuclear gaze palsy (1) [5,8,18]. 

### 2.6. Episodic Ataxia and Other Paroxysmal Motor and Non-Motor Disorders

EA is a clinically heterogeneous disorder characterized by recurrent spells of ataxia lasting minutes to hours [15] which has been associated over time to a number of genes besides *CACNA1A* and *KCNA1* (*CACNB4*, *SLC1A3*, *FGF14*, *SLC2A1*, *ATP1A3*, *PRRT2*) accounting for the majority of cases [15,29]. To date, two patients with two different *KCND3* mutations have been reported with EA as the main disease feature [15,20]. The first patient suffered since the age of 30 of paroxysmal episodes of vertigo and gait ataxia [20]. During the interictal period, his neurological examination showed pyramidal signs and saccadic ocular pursuit. Of note, brain MRI disclosed a cerebellar vermian atrophy. The second is a 17-year-old patient presenting with paroxysmal spells of ataxia, vertigo, dysarthria, and earfulness, lasting from minutes to hours [15]. Interictal signs include only a gaze-evoked nystagmus [15].

A further patient reported by Lee and colleagues presented at the age of 30 years with intermittent diplopia and later suffered from progressive gait ataxia and intermittent cerebellar dysarthria [5]. 

### 2.7. Cognition and Behavioral Disorders

Cognitive impairment and neurodevelopmental disorders such as developmental delay (DD), ID and learning disabilities are variably present in patients with *KCND3* mutations. Cognitive impairment was reported in four out of the nine members of the first family described [9]. In the late-onset cohort, cognitive impairment was usually mild and present in almost 25% of patients [9,16,18]. Formal neuropsychological evaluation revealed executive and visuospatial functions as the most severe deficits [9,16,18]. Notably, in the early-onset cohort, neurodevelopmental disorders and/or subsequent cognitive impairment were described in all patients reported, surpassing even ataxia in the frequency rating of symptoms. Psychiatric symptoms were reported in three different families [5,9,16]. Depression was present in nine patients, ranging from mild to severe [9,16]. Anxiety, obsessive compulsive disorder, delusional thoughts, and mild aggressiveness were also reported [5,16]. 

### 2.8. Movement Disorders

Movement disorders (MD) were present in 23 out of 68 patients. Myoclonus occurred in five cases, involving limbs, trunk, and abdomen [6,9,16,17]. Multifocal dystonia mainly affecting upper and/or lower limbs was reported in three early-onset patients [6,17]. One of them suffered from blepharospasm [6]. Trunk tremor was present in one patient [18] and a head tremor was described in two patients [9,18] and anticipated the onset of a frank ataxic syndrome. Holmes tremor was described in two cases [9]. Intentional tremor was present in one patient [9] and an unspecified extra-pyramidal tremor in another two [5]. Parkinsonian features, including rigidity or bradykinesia, were the most frequent movement disorder in *KCND3* deficient patients, occurring in 9 out of 53 late-onset patients and in 3 out of 15 early-onset patients [5,6,16,18]. 

### 2.9. Epilepsy

Epilepsy was reported in 7 out of the 15 patients with disease onset during infancy or childhood [13,16,19], often as presenting symptom (5/7) [16,19]. Partial or generalized seizures were usually well controlled with pharmacological monotherapy [13,16]. Drug-resistant epilepsy was reported in a single patient with epileptic encephalopathy [19].

EEG abnormalities included: theta waves in the frontal or in both the frontal and the parietal area [13,16]; spike-wave or poly-spike wave in frontal or frontotemporal area [16]; bursts of high amplitude sharp wave during photic stimulation and hyperventilation (in a patient who never experienced seizures [17]); and epileptic abnormalities such as multifocal spikes, spike-wave, and slow wave prominent in the right temporal lobe in the subject with epileptic encephalopathy [19]. 

### 2.10. Other Features: Pyramidal Signs and Peripheral Neuropathy 

Pyramidal signs including hyperreflexia, ankle clonus, spasticity, and extensor plantar reflex were observed in a minority of patients in both early and late onset group (16/68) [5,8,9,12,16,17,20]. 

Peripheral neuropathy was observed only in the late-onset group (five patients) [5,9,18]. Both sensory and sensorimotor axonal polyneuropathy were described, but electrophysiological tests were not systematically performed; therefore, their real prevalence may have been underestimated.

### 2.11. Neuroimaging 

Cerebellar atrophy was the main finding on brain MRI in 30/35 patients [5,6,8,9,14,16,17,18,20]. Combined atrophy of cerebellar hemispheres and vermis was present in about half of the cases [5,6,9,14,17], while isolated vermis atrophy was described in 11 cases [5,8,16,18,20]. Two patients showed hemispheric cerebellar atrophy without vermian involvement [9]. Cerebral atrophy and white matter lesions were reported in association with cerebellar atrophy in a minority of patients [6,9,17,18]. Notably, five patients with an early-onset form had a normal MRI [9,13,16,19]. 

Neuroradiological findings of 35 literature patients that underwent brain MRI are summarized in Table 2. 

### 2.12. Genotype

To date, 15 different *KCND3* pathogenetic variants associated with a neurological phenotype have been reported. Among recurrent variants, p.Phe227del was found in four unrelated families [5,16], while p.Gly345Val, p.Thr377Met, and p.Ser390Asn were each carried by two families, [5,8,14,18]. 

Variants reported as pathogenic affect residues located from the first Kv4.3 extracellular loop (p.Lys214Arg) [20] to the sixth transmembrane domain (p.Leu450Phe) [12]. A mutational hotspot seems to be located between the pore loop and the S6 domains (8/17) [5,6,8,9,14,17,18,19].

Electrophysiological and immunohistochemical studies revealed a loss of function pathogenic mechanism leading to aberrant Kv4.3 functioning through different pathways, such as impaired protein trafficking, reduced outward K^+^ current, and increased protein degradation [5,6,8,13,18]. Genotype data were summarized in Figure 3 and in Table 3.

Two *KCND3* gain of function variants previously associated with sudden unexplained death and cardiological syndrome (p.Leu450Phe and p.Val392Ile) [12,19], were described respectively in a patient with late-onset cerebellar ataxia [12] and in a patient with epileptic encephalopathy [19], with no evidence of cardiological issues. 

## 3. Discussion

*KCND3* gene deficiency, associated to SCA19/22 [5,8], is a complex neurological syndrome encompassing a still expanding spectrum of neurological features beside ataxia. Oculomotor disturbances, DD later evolving to ID, cognitive decline, movement disorder, epilepsy, pyramidal signs, neuropathy, EA, and other paroxysmal neurological disturbances were added to the clinical spectrum of this condition over time, as detailed in this review. 

The pathophysiology underlying this wide spectrum of manifestations is not fully understood.

Based on our literature review, we propose two main clinical phenotypes, according to the age at onset.

Patients with early onset forms present with neurodevelopmental disorder or epilepsy preceding the onset of cerebellar signs. Movement disorders can emerge later in the disease course in half of them. 

Late-onset patients show a pure slowly progressive cerebellar ataxia with possible cognitive decline, peripheral neuropathy, parkinsonism, and other movement disorders.

Such a different age-dependent clinical trajectory was observed in few other genetic neurological disorders presenting with ataxia, including *PUM1* deficiency and a further Kvs disease due to [30] *KCNC3* gene mutation (SCA type 13) [31]. 

Inherited neurological channelopathies are known to be causative of a great number of paroxysmal and non-paroxysmal neurological disorders [32,33].

Our literature review identified two patients with different *KCND3* variants presenting with EA associated to interictal ocular motility abnormalities or pyramidal signs [15,20]. Due to the limited follow-up of these patients, it is uncertain if in a subset of *KCND3* patients, EA may anticipate the development of a progressive ataxia as seen in other channelopathies [33]. 

Our patient showed paroxysmal neurological symptoms in a disease stage when fixed ataxic signs were not yet obvious. Further studies are needed in order to clarify the prevalence, phenomenology, and pathogenic mechanisms of EA and other paroxysmal neurological disorders in *KCND3* defects.

Movement disorders in *KCND3* patients are heterogenous and their prevalence is relatively high. A single post-mortem study revealed normal basal ganglia, except for a pallor of substantia nigra [34], and PET studies failed to detect any alterations in basal ganglia metabolism [18]. There is increasing evidence and interest regarding to the cross-talk between cerebellum and basal ganglia [35]. The cerebellum is currently considered to play a major role in both dystonia and parkinsonism, possibly because of its temporal processing function integrated with basal ganglia activity [36,37,38]. This essential function depends on a complex balancing of cerebellar microcircuit, synaptic receptors, neurotransmitters release, and ionic channels [38]. Clinical insights come from the presence of movement disorders in patients with both acquired or inherited cerebellar issues [39,40,41,42]. Parkinsonism is a prominent feature of different SCAs (SCA 2, 3, 6, 8, and 17) [43] including SCA 19/22.

Co-occurrence of dystonia and ataxia, named dystonia-ataxia syndrome, can be caused by more than 100 different genes [41]. Our patient showed a longstanding dystonic syndrome with prominent cervical dystonia before the emergence of ataxic features with cerebellar atrophy. While cervical dystonia can precede the onset of ataxia in a number of other genetic ataxic syndromes, such as SCA2 [44], SCA 3 [45], ataxia-teleangectasia [46], and ataxia associated to vitamin E deficiency [47], the pathophysiological explanation of this fascinating association has been poorly explored in the literature so far.

The existence of a neural integrator for the control of head movements, analogous of that in the ocular motor system leading to centripetal drift of the eyes and consequent gaze-evoked nystagmus was proposed by Klier and Colleagues in 2002 [48]. Recently, a counterpart of gaze-evoked nystagmus was identified for head movements, suggesting a novel pathophysiological explanation for cervical dystonia as the result of dysfunction of a head neural integrator, which may be impaired as a consequence of an altered cerebellar, basal ganglia, or peripheral input [49]. In a large series of subjects with cervical dystonia, MRI studies demonstrated a cerebellar involvement in 14% of patients (i.e., atrophy, tumor, infarct, cyst, white matter hyperintensities, or ectopia) [50].

A role for the cerebellum in cognitive and affective functions was first pointed out by the description in adult patients of the cerebellar cognitive affective syndrome (CCAS) [51]. CCAS is characterized by defects in executive functions, spatial cognition, linguistic processing, and affect regulation [52]. Both psychiatric and cognitive issues in late-onset *KCND3* patients are consistent with CCAS. 

Later on, the importance of the cerebellum in neurodevelopmental trajectories was studied in patients with congenital or early acquired cerebellar disorders [53,54]. The strikingly high percentage of DD, ID, and learning disabilities in *KCND3* early-onset patients remarks this concept. 

Cognitive decline and ID are non-motor symptoms described in several types of SCAs [55,56]. A recent work attempted to predict type and severity of cognitive issues in the expanded-polyglutamine tracts SCAs according to the underlying genetic diagnosis (i.e., SCA type 1, 2, 3, 6, 7, 8, and 17 and dentato pallidoluysian atrophy) [57]. A prominent executive dysfunction was found in SCA 1 subjects compared to SCA type 2 and 3 [58]. Cognitive decline rate was associated to the severity of motor degeneration in SCA type 1, 2, 3, and 6 but not in SCA type 7 [59]. 

ID was associated to SCA 13 (due to *KCNC3* defect) [55,56] and, according to the present review, to *KCND3* defect (early-onset presentation). 

The association between SCAs and epilepsy is best known in SCA type 10 and 13 [16,60]; however, sporadic individuals suffering from epilepsy associated with different types of SCAs have been reported [60,61]. 

Epilepsy in SCA type 10 develops some years after the onset of ataxia [16]. In SCA 13, it is more frequent in younger patients [62]. Notably, epilepsy in SCA 19/22 deficiency shares similarities with epilepsy in SCA type 13. Both these SCA subtypes are caused by a Kv defect [62], known to be related to inherited human epilepsy, even to severe epileptic phenotypes [63,64]. 

Brain MRI abnormalities include isolated vermis atrophy, isolated cerebellar hemispheric atrophy, or global cerebellar atrophy and possibly white matter lesions and cerebral atrophy. Interestingly, a number of early-onset patients had a normal brain MRI, with four of them showing ataxia at time of examination [9,13,16]. In our patient, cerebellar atrophy was first seen at the age of 23 and was a late disease finding. 

## 4. Methods and Materials

### 4.1. Case Report

We reported clinical information, genetic data, neuroimaging, and video recordings of a new patient with a novel *KCND3* variant. Written informed consent was obtained in accordance with institutional review board regulations and protocols to disclose clinical information, neuroimaging, and video recordings. 

### 4.2. Molecular Studies 

CES was performed using the Illumina Nextera Trusight One capture kit on the Illumina Miseq Dx platform (Illumina, San Diego, CA, USA) with a percentage of aligned sequences: 99.7%; mean region coverage depth: 95.5×; regions covered at least 20×: 91.3%; percentage of bases with Q ≥ 30: 95.9%; data referring to the index patient.

The sequencing product was aligned to the reference genome (GRCh37-hg19). The variants call, filtering, and prioritization were performed using an in-house pipeline and different tools, including Variant Interpreter (Illumina), Phenomizer browser (http://compbio.charite.de/phenomizer/, last access date: 11 June 2020), and Decipher browser (https://decipher.sanger.ac.uk/, last access date: 11 June 2020). Variants were filtered for: consequence (missense, nonsense, frame shift, inframe indels and essential splice variants, SIFT prediction as deleterious and not tolerated, Polyphen prediction as probably and possibly damaging), metrix (filter PASS), and population frequency (GnomAD, GMAF and 1000Genomes frequencies < 0.01). Variants analysis disclosed a never reported missense variant in *KCDN3* gene. 

The variant, validated by Sanger sequencing in the index patient and his parents, was confirmed de novo (Figure 1). The candidate variant in *KCDN3* gene was tested by different bioinformatic tools such as Mutation Taster [22], Predict SNP [23], HOPE [24], and VarSome [25].

### 4.3. Literature Review

A comprehensive search of the medical literature (PubMed, Medline, Cochrane CENTRAL, Google Scholar) was conducted to identify all English-language papers reporting *KCND3* mutation carriers. 

“KCND3 deficiency”, “SCA19”, “SCA22”, and “SCA19/22” were used as search terms [5,6,8,9,10,12,13,14,15,16,17,18,19,20]. We reviewed 86 published cases from 14 papers and selected 68 cases from 13 papers [5,6,8,9,12,13,14,15,16,17,18,19,20], for which sufficient clinical data about clinical presentation and evolution were available. 

Patients were classified according to the age of onset in two main subgroups: infantile-childhood onset group (from now on called early-onset group), including 15 patients, and a juvenile-adult onset group (from now on called late-onset group), including 53 patients. Given the partial overlap between childhood and adolescence in terms of age range [65] and the presence of distinctive clinical patterns of disease onset and evolution in patients with less than 12 years and in patients with more than 12 years of age, patients with more than 12 years were included in the late-onset group.

Detailed clinical, radiological, and genetic data are summarized in Table 1, Table 2 and Table 3. Extended clinical data are available in Appendix A.

## 5. Conclusions

In summary, we reported a new patient with *KCND3* mutation with neurodevelopmental disorder and prominent dystonia, with a clinical course paradigmatic of the early onset forms. Although the pathophysiological underpinnings were not elucidated, based on our literature review, presentation patterns and disease course differ between early and late onset forms, thus delineating two well differentiated phenotypic trajectories according to the age of onset.

## Figures and Tables

**Figure 1 ijms-21-05802-f001:**
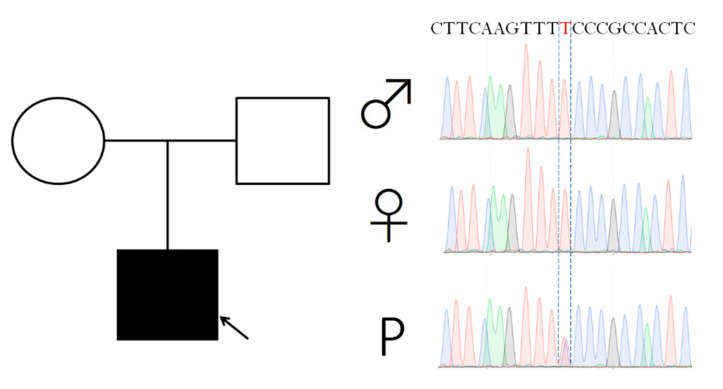
Pedigree and *KCND3* Sanger sequencing of the index patient. Patient is pointed by an arrow. P: patient.

**Figure 2 ijms-21-05802-f002:**
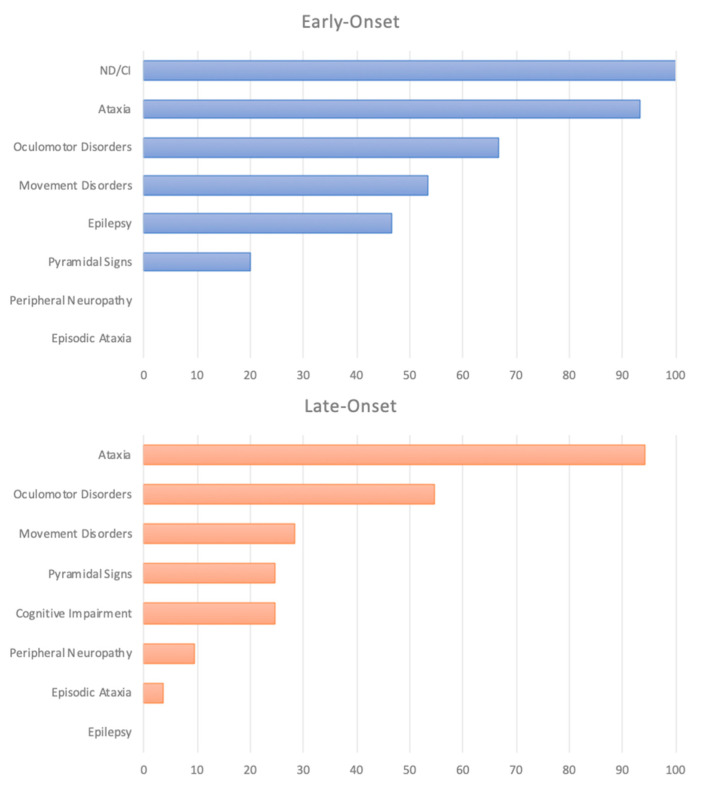
Percentage of symptoms in the early and late-onset cohort. ND: Neurodevelopmental disorders CI: Cognitive impairment.

**Figure 3 ijms-21-05802-f003:**
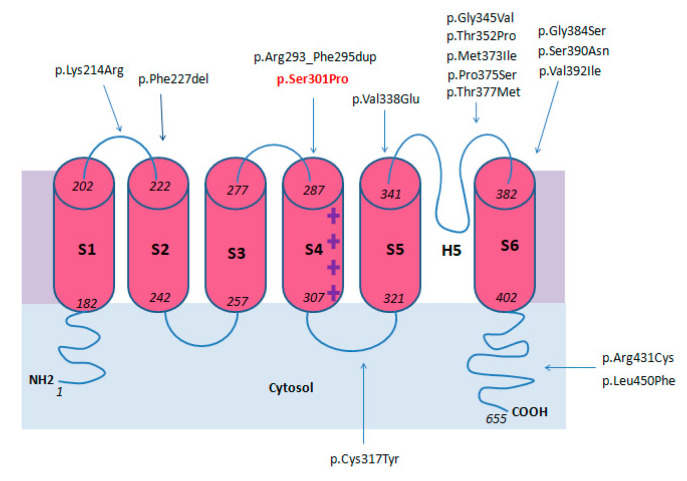
Locations of reported variants in KCND3 domains.

**Table 1 ijms-21-05802-t001:** Clinical data of 68 patients *KCND3* mutation carriers.

Age of Onset	28 (1–90)
First symptom	Ataxia (42)
	Neurodevelopmental disorders/cognitive impairment (5)
	Epilepsy (5)
	Episodic ataxia (2)
	Head tremor (2)
	Intermittent diplopia (1)
	Psychiatric symptoms (1)
Ataxia	64/68
Episodic ataxia	2/68
Neurodevelopmental disorders/Cognitive impairment	28/68
Movement disorder	23/68
Parkinsonism	12
Tremor	7
Myoclonus	5
Dystonia	3
Epilepsy	7/68
Focal	3
Generalized	2
Mixed	1
Unspecified	1
Pyramidal signs	16/68
Oculomotor disorders	39/68
Saccadic Pursuit	27
Nystagmus	22
GEN	16
Unspecified	11
Downbeat Nystagmus	1
Dysmetric saccades	3
Vertical ophtalmoplegia	1
Supranuclear palsy	1
Slow saccades	1
Diplopia/Intermittent Diplopia	2
Peripheral Neuropathy	5/68

**Table 2 ijms-21-05802-t002:** MRI findings of 35 *KCND3* mutation carriers.

	Patients (n./35)	References
**Normal**	5	[9,13,16,19]
Global cerebellar atrophy	17	[5,6,9,14,17]
Hemispheric cerebellar atrophy	2	[9]
Vermian cerebellar atrophy	11	[5,8,16,18,20]
Cerebral atrophy	5	[6,9,17]
White matter lesions	3	[9,18]

**Table 3 ijms-21-05802-t003:** Genotype data of *KCND3* pathogenetic variants reported in literature.

Variant	N of Different Families	Location	Pathogenetic Mechanism	References
Phe227del rs397515475	4	S2 domain	Impaired protein trafficking	[5,16]
Gly345Val rs797045634	2	Pore loop	ND	[5]
Thr377Met rs1571636501	2	Pore loop	Increased protein degradation Impaired protein trafficking Reduced outward K^+^ current	[5,6,18]
Ser390Asn rs397515478	2	S6 domain	Increased protein degradation Impaired protein trafficking Reduced outward K^+^ current	[8,14]
Met373Ile rs397515477	1	Pore loop	Increased protein degradation Impaired protein trafficking Reduced outward K^+^ current	[8]
Thr352Pro rs397515476	1	Pore loop	Increased protein degradation Impaired protein trafficking Reduced outward K^+^ current	[8]
Val338Glu rs1571939827	1	S5 Domain	Increased protein degradation Reduced outward K^+^ current	[5,6]
Leu450Phe rs150401343	1	C-terminal tail	Gain of function	[12]
Arg293_295PheDup No rsID	1	S4 domain	Reduced outward K^+^ current function	[13]
Arg431Cys rs777183510	1	C-terminal tail	ND	[15]
Gly384Ser No rsID	1	S6 domain	ND	[17]
Val392Ile rs786205867	1	S6 domain	Gain of function	[19]
Lys214Arg rs142744204	1	First extracellular loop	ND	[20]
Cys317Tyr rs1571939905	1	Second intracellular loop	Increased protein degradation Reduced outward K^+^ current	[6]
Pro375Ser rs1571636508	1	Pore loop	Increased protein degradation Impaired protein trafficking Reduced outward K^+^ current	[6]
Ser301Pro rs79821338	1	S4 domain	ND	Current report

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
