# Peer review of "KCND3-Related Neurological Disorders: From Old to Emerging Clinical Phenotypes"

_ijms, 2020, doi:10.3390/ijms21165802_

Round 1

Reviewer 1 Report

1. IRB approval number should be included in the material part for reference.
2. This variation (NM_172198.2:c.901T>C) already has a record in dbSNP rs79821338. This accession should be noted in the abstract and main text for better reference.
3. The authors described 2 NGS panels failed to identify the potential causative variation, could describe the gene list in these 2 panel?
4. In Section 2.2, the authors mentioned they preformed Sanger sequencing on the proband and his parents at the identified variation site, but did not present the corresponding results. Including a pedigree as a figure could further clarify the genetic/phenotype correlation
5. In Table 2 and supplementary Table, each variation is suggested to include a public accession for searching of extensive information, e.g. dbSNP rsID.

Author Response

Response to Reviewer 1 Comments

Point 1: IRB approval number should be included in the material part for reference.

Response 1: According to our Ethical Committee, Case Report publication does not require a specific approval, but the consent of patient (or tutor), which is stored in the medical records and available upon request.

Point 2: This variation (NM_172198.2:c.901T>C) already has a record in dbSNP rs79821338. This accession should be noted in the abstract and main text for better reference.

Response 2: We thank the reviewer for this suggestion. We have added a sentence in the Abstract reporting the rs number (line 23) and we have modified a sentence in the Results section (lines 92-95).

Point 3: The authors described 2 NGS panels failed to identify the potential causative variation, could describe the gene list in these 2 panel?

Response 3: The NGS panel for genetic epilepsy includes 147 different genes (AARS, ADRA2B, ALDH7A1, ALG13, AP3B2, ARHGEF9, ARV1, ARX, ATP1A2, ATP1A3, ATRX, BRAT1, CACNA1A, CACNB4, CAD, CDKL5, CHD2, CHRNA2, CHRNA4, CHRNB2, CPA6, CSKN1G1, CSTB, DDX3X, DENND5A, DEPDC5, DLAT, DNM1, DOCK7, EEF1A2, EPM2A, FARS2, FGF12, FOLR1, FOXG1, FOXP1, FOXP2, FRRS1L, GABRA1, GABRB3, GABRG2, GLRA1, GLRB, GNAO1, GNB1, GOSR2, GRIN1, GRIN2A, GRIN2B, GRIN2D, HACE1, HCN1, HNRNPU, HUWE1, IQSEC2, ITPA, ITRP1, KANSL1, KCNA1, KCNA2, KCNB1, KCNC1, KCHN1, KCNJ10, KCNK18, KCNMA1, KCNQ2, KCNQ3, KCNT1, KCTD7, LGI1, LIAS, MBD5, MDH2, MECP2, MEF2C, MTHFR, MTOR, NECAP1, CHLRC1, NPRL2, NPRL3, NRXN1, PC, PCHD19, PDHA1, PDHB, PDP1, PIGA, PIGG, PIGN, PIGQ, PIGT, PLCB1, PLPBP, PNKD, PNKP, PNPO, POLG, PRICKLE1, PRRT2, PURA, QARS, RELN, ROGDI, SCARB2, SCN1A, SCN1B, SCN2A, SCN8A, SCN10A, SERPINI1, SHANK3, SIK1, SLC12A5, SLC13A5, SLC19A3, SLC1A2, SLC1A3, SLC25A12, SLC25A22, SLC2A1, SLC35A2, SLC35A3, SLC6A1, SLC6A5, SLC6A8, SLC9A6, SMC1A, SON, SPATA5, SPTAN1, ST3GAL3, ST3GAL5, STX1B, STXBP1, SYN1, SYNGAP1, SYNJ1, SZT2, TBC1D24, TCF4, TPP1, UBA5, UBE3A, WDR45, WWOX).

The NGS panel for genetic movement disorders includes 28 genes (TOR1A, THAP1, PRKPA, TAF1, TIMM8A, GCH1, TH, SGCE, ATP1A3, PNKD, PRRT2, SLC2A1, CIZ1, ANO3, GNAL, NKX2.1, ATM, HPCA, COL6A3, KCTD17, TOR1AIP1, ADCY5, GNAO1, PNKP, CACNA1A, GRIN1, KTM2B, SYT).

Point 4: In Section 2.2, the authors mentioned they performed Sanger sequencing on the proband and his parents at the identified variation site, but did not present the corresponding results. Including a pedigree as a figure could further clarify the genetic/phenotype correlation

Response 4: We have added a figure (Figure 1) to clarify the genotype/phenotype correlation (line 93) and modified the text in Methods and materials section (lines 343-344). We also have specified in the abstract (line 22) that the variant was de novo.

Point 5: In Table 2 and supplementary Table, each variation is suggested to include a public accession for searching of extensive information, e.g. dbSNP rsID.

Response 5: We thank the reviewer for this suggestion. When available, we included a public accession number (rs ID) in both table 3 and supplementary table for extensive information.

Here are few clarifications regarding some inconsistencies we have encountered:

  • Lee and Colleagues reported a c.1013T>C (p.Val338Glu) variant (family D). In dbSNP the associated rsID should be rs1571939827. However, in dbSNP the variant associated to this rsID has a different nucleotide change (c.1013T>A p.Val338Glu). Given that c.1013T>C should not result in p.Val338Glu substitution, we have associated the rs1571939827 ID to Lee variant  and changed c.1013T>C with  c.1013T>A in the supplemental table .
  • We have associated to c.679_681delTTC (p.Phe227del) variant (Lee family A and B, Huin family A and B) a rs397515475 ID from dbSNP. Although this rsID is extracted from the same paper by Lee and Colleagues, the described variant is associated with a different nucleotidic change (c.680_682delTCT).

Reviewer 2 Report

This is a well-done report on KCND3/K4.3. The authors provide a fair review of the literature, adding also a case description.

Minor comments:

-why did the authors checked only PubMed and not another database? 

-please explain better why 12 years is selected as a cut-off. A statistical justification is required

-dystonia may result from an error in the timing. Cerebellum plays a key-function in timing. Mutations in potassium channels will impact on the cerebellar circuitry. See  https://pubmed.ncbi.nlm.nih.gov/30259343/

Author Response

Response to Reviewer 2 Comments

Point 1: Why did the authors checked only PubMed and not another database? 

Response 1: We thank Reviewer#2 for this comment. We have also checked  Cochrane CENTRAL, Medline, and Google Scholar and we didn’t find further papers reporting detailed data about the neurological phenotype and clinical course of KCND3 mutations carriers, which was the aim of the present review. We have added a sentence to the text to acknowledge this (lines 347-348) and we have rearranged the paragraph to clarify our search strategy (lines 349-351).

Point 2: Please explain better why 12 years is selected as a cut-off. A statistical justification is required

Response 2: We thank the reviewer for this comment. We aimed to stratify patients in 2 different groups according to the age of onset (Early-onset: infancy to childhood; Late-onset: adolescence to adulthood). However, there is no a clear cut-off between childhood and adolescent in terms of age, especially in the age range 10-15 [Sawyer, S. M.; Azzopardi, P. S.; Wickremarathne, D.; Patton, G. C. The Age of Adolescence. The Lancet Child & Adolescent Health 2018, 2 (3), 223–228. https://doi.org/10.1016/S2352-4642(18)30022-1.]. In this context, reviewing the clinical history and course of literature patients, we have chosen 12 years of age as a reasonable cut-off. We have changed a sentence in Methods and materials section (lines 352-357) and we have added the above cited paper to References (reference 65) to clarify this criterion.

Point 3: Dystonia may result from an error in the timing. Cerebellum plays a key-function in timing. Mutations in potassium channels will impact on the cerebellar circuitry. See https://pubmed.ncbi.nlm.nih.gov/30259343/

Response 3: We thank the reviewer for this comment and suggested reference. Indeed, this interesting consensus paper further expands and clarifies the relationship between cerebellar dysfunction and movement disorder. We have added a sentence to Discussion (line 275-277) and the reference of the suggested paper (ref. 38).

Round 2

Reviewer 1 Report

All raised questions were answered satisfactory.